# Unsuppressed HIV infection impairs T cell responses to SARS-CoV-2 infection and abrogates T cell cross-recognition

**Thandeka Nkosi[1†], Caroline Chasara[1†], Andrea O Papadopoulos[1]\*, Tiza L Nguni[1], Farina Karim[1], Mahomed-Yunus S Moosa[2], Inbal Gazy[3], Kondwani Jambo[4,5], COMMIT-KZN-Team[1], Willem Hanekom[1,6], Alex Sigal[1], Zaza M Ndhlovu[1,2,7]\***

[1]Africa Health Research Institute, Nelson R. Mandela School of Medicine, University of Kwa-Zulu Natal, Durban, South Africa; [2]HIV Pathogenesis Program, School of Laboratory Medicine and Medical Sciences, University of KwaZulu-Natal, Durban, South Africa; [3]KwaZulu-Natal Research Innovation and Sequencing Platform (KRISP), Nelson R Mandela School of Medicine, University of KwaZulu-Natal, Durban, South Africa; [4]Malawi-Liverpool-Wellcome Trust Clinical Research Programme, Blantyre, Malawi; [5]Liverpool School of Tropical Medicine, Liverpool, United Kingdom; [6]Division of Infection and Immunity, University College London, London, United Kingdom; [7]Ragon Institute of MGH, MIT and Harvard, Cambridge, United States

**\*For correspondence:**
andrea.papadopoulos@ahri.org (AOP);
zndhlovu@mgh.harvard.edu (ZMN)

[†]These authors contributed equally to this work

**Abstract** In some instances, unsuppressed HIV has been associated with severe COVID-19 disease, but the mechanisms underpinning this susceptibility are still unclear. Here, we assessed the impact of HIV infection on the quality and epitope specificity of SARS-CoV-2 T cell responses in the first wave and second wave of the COVID-19 epidemic in South Africa. Flow cytometry was used to measure T cell responses following peripheral blood mononuclear cell stimulation with SARS-CoV-2 peptide pools. Culture expansion was used to determine T cell immunodominance hierarchies and to assess potential SARS-CoV-2 escape from T cell recognition. HIV-seronegative individuals had significantly greater CD4[+] T cell responses against the Spike protein compared to the viremic people living with HIV (PLWH). Absolute CD4 count correlated positively with SARS-CoV-2-specific CD4[+] and CD8[+] T cell responses (CD4 r=0.5, p=0.03; CD8 r=0.5, p=0.001), whereas T cell activation was negatively correlated with CD4[+] T cell responses (CD4 r=−0.7, p=0.04). There was diminished T cell cross-recognition between the two waves, which was more pronounced in individuals with unsuppressed HIV infection. Importantly, we identify four mutations in the Beta variant that resulted in abrogation of T cell recognition. Taken together, we show that unsuppressed HIV infection markedly impairs T cell responses to SARS-Cov-2 infection and diminishes T cell cross-recognition. These findings may partly explain the increased susceptibility of PLWH to severe COVID-19 and also highlights their vulnerability to emerging SARS-CoV-2 variants of concern.

## Editor's evaluation

This paper provides important descriptive evidence that untreated HIV infection has important negative effects on T cell responses to SARS-CoV-2, particularly in regards to cross recognition of new variants. Treatment of HIV with ART appears to partially reverse suppression of SARS-CoV-2 specific cellular immune responses.

## Introduction

Despite measures to contain the spread of SARS-CoV-2 infection, the pandemic is persisting, with a devastating impact on healthcare systems and the world economy (*Verma et al., 2021*). The research community rapidly mobilized and developed vaccines and therapeutics at unprecedented speed (*Polack et al., 2020*; *Ball, 2021*). COVID-19 vaccines have prevented serious illness and death and have in some cases interrupted chains of transmission at community level (*Kampf, 2021*). However, the COVID-19 pandemic remains a major concern in Africa due to dismal vaccine coverage (*WHO, 2021a*) and the emergence of variants of concern that may be more transmissible, cause more severe illness, or have the potential to evade immunity from prior infection or vaccination (*Shinde et al., 2021*).

The interaction of HIV-1 infection, common in sub-Saharan Africa (*H. The Lancet, 2020*), with COVID-19 remains understudied. Initial small studies reported that people living with HIV (PLWH) had similar or better COVID-19 outcomes (*Calza et al., 2020*; *Lee et al., 2021*). Larger epidemiological studies have demonstrated increased hospitalization and higher rates of COVID-19-related deaths among PLWH compared with HIV-negative individuals (*Davies, 2020a*; *Geretti et al., 2020*; *Bhaskaran et al., 2021*; *Vizcarra et al., 2020*). Other studies have linked HIV-mediated CD4$^+$ T cell depletion to suboptimal T cell and humoral immune responses to SARS-CoV-2 (*Riou, 2021*). A recent study showed prolonged shedding of high titer SARS-CoV-2 and emergence of multiple mutations in an individual with advanced HIV and antiretroviral treatment (ART) failure (*Karim et al., 2021b*).

Although B cells have repeatedly been shown to play a pivotal role in immune protection against SARS-CoV-2 infection and antibody responses and are typically used to evaluate immune responses to currently licensed COVID-19 vaccines (*Sahin et al., 2020*; *Khoury et al., 2021*), mounting evidence suggest that T cell responses are equally important. For instance, strong SARS-CoV-2-specific T cell responses are associated with milder disease (*Riou, 2021*; *Sette and Crotty, 2021*; *Liao et al., 2020*; *Schub et al., 2020*; *Rydyznski Moderbacher et al., 2020*). Moreover, T cell responses can confer protection even in the absence of humoral responses, given that patients with inherited B cell deficiencies or hematological malignancies are able to fully recover from SARS-CoV-2 infection (*Bange et al., 2021*). In some instances, COVID-19 disease severity has been attributed to poor SARS-CoV-2-specific CD4$^+$ T cell polyfunctionality potential, reduced proliferation capacity, and enhanced HLA-DR expression (*Riou, 2021*). Importantly, a recent study identified nonsynonymous mutations in known MHC-1-restricted CD8$^+$ T cell epitopes following deep sequencing of SARS-CoV-2 viral isolates from patients, demonstrating the capacity of SARS-CoV-2 to escape from CTL recognition (*Agerer et al., 2021*). Regarding vaccine-induced T cell responses, it was recently shown that mRNA vaccines can stimulate Th1 and Th2 CD4$^+$ T cell responses that correlate with post-boost CD8$^+$ T cell responses and neutralizing antibodies (*Painter et al., 2021*). The cited examples, herein, highlight the need to gain more insight into T cell-mediated protection against COVID-19 (*Altmann and Boyton, 2020*).

This study used a cohort of PLWH and HIV-seronegative individuals diagnosed with COVID-19 during the first wave dominated by the wild-type (wt) D614G virus (*Tegally et al., 2021c*), and the second wave dominated by the Beta variant. Peripheral blood mononuclear cells (PBMCs) were used to determine the impact of HIV infection on SARS-CoV-2-specific T cell responses and to assess T cell cross-recognition. Our data showed impaired SARS-CoV-2-specific T cell responses in individuals with unsuppressed HIV infection and highlighted poor cellular cross-recognition between variants, which was more pronounced than those with unsuppressed HIV. The muted responses in unsuppressed HIV infection may be attributable to low absolute CD4 count and immune activation. Importantly, we identified mutations in the Beta variant that could potentially reduce T cell recognition. Taken together, these data highlight the need to ensure uninterrupted access to ART for PLWH during the COVID-19 pandemic.

## Results

Study participants were drawn from a longitudinal observational cohort study that enrolled and tracked patients with a positive COVID-19 qPCR test presenting at three hospitals in the greater Durban area. Study participants were recruited into this study based on HIV status and sample availability. They include 25 participants recruited during the first wave (wt) of the pandemic in KwaZulu-Natal from June to December 2020 (*Karim et al., 2021a*). Twenty-three second wave (Beta variant) participants were

**Table 1.** Donor characteristics stratified by HIV status.

| | All (n=48) | HIV-neg (N=17) | HIV+suppressed (n=17) | HIV+viremics (N=14) | Statistics |
|---|---|---|---|---|---|
| **Demographics** | | | | | |
| Age years, median (IQR) | 40.5 (30–51.75) | 45 (27–53.5) | 45 (39.5–54) | 31.5 (26.5–42) | 0.036* (KW) |
| Male sex, n (%) | 14 (29.16) | 8 (47.05) | 3 (17.64) | 3 (21.42) | 0.2 (0.82–10) (F) |
| **HIV-associated parameters** | | | | | |
| HIV viral load copies/ml | | | | 19,969 (2335–43,568) | |
| CD4 cells/µl median (IQR) | 661 (398.5–836.5) | 834.5 (739.3–1029) | 661 (494–789.5) | 301 (113.8–568) | 0.0002** (KW) |
| **Disease severity** | | | | | |
| Asymptomatic, n (%) | 9 (18.75) | 4 (23.52) | 3 (17.64) | 2 (14.28) | 0.6 (0.32–9.53) (F) |
| Mild | 29 (60.42) | 12 (70.59) | 10 (58.82) | 7 (50) | 0.01* (0.13–0.84) (F) |
| Severe/oxygen supplementation | 8 (16.67) | 1 (5.88) | 4 (23.52) | 3 (21–42) | 0.33 (0.48–49.67) (F) |
| Death, n (%) | 1 (2.1) | 0 | 0 | 1 (7.1) | 0.46 (F) |

P values calculated by Kruskal-Wallis test for unpaired three groups (KW) or Fischer's exact test (F).

recruited from January to June 2021. All study participants were unvaccinated because the COVID-19 vaccine was not readily available in South Africa at the time. Study participants were stratified into three groups, namely HIV-seronegative (HIV-neg), people living with HIV (PLWH) with viral load below 50 copies/ml, here termed (suppressed), and PLWH with detectable viral load of ≥1000 copies/ml (viremic). Study participants included HIV-seronegative (HIV-neg) (n=17). PLWHs (n=31) were subdivided into suppressed (n=17) and viremic (n=14). The male-to-female ratio and age distribution were comparable between PLWH and HIV-seronegative groups (*Table 1*). The median CD4 count for PLWH (suppressed 661 and viremic 301) (p=0.0002, *Table 1*). Study participants had predominantly mild COVID-19 disease that did not require supplemental oxygen or ventilation (*Table 1*).

## Unsuppressed HIV infection is associated with altered SARS-CoV-2-specific CD4+ and CD8+ T cell responses

Immunity to SARS-CoV-2 typically induces robust T cell responses, but the impact of HIV infection on these responses has not been fully elucidated (*Bange et al., 2021*; *Riou et al., 2021*; *Grifoni et al., 2020*). Thus, we sought to determine the impact of HIV infection on SARS-CoV-2-specific CD4+ and CD8+ T cell responses. PBMCs were stimulated with PepTivater 15 mer megapools purchased from Miltenyi Biotec. The pools contained predicted CD4 and CD8 epitopes spanning the entire Spike coding sequence (aa5-1273). Intracellular cytokine staining of peptide-stimulated PBMCs was followed by flowcytometric analyses described in the Materials and methods section. The samples used for these analyses were collected between 2 and 4 weeks after COVID-19 PCR positive diagnosis. The time points were selected based on longitudinal T cell analysis that showed SARS-CoV-2-specific T cell responses peaked between 14 and 30 days after PCR positive diagnosis (data not shown), consistent with other studies (*Keeton et al., 2021*; *Keeton et al., 2022*). Representative flow plots for each group and aggregate data show viremic PLWH had significantly lower frequencies of SARS-CoV-2-specific IFN-γ/TNF-α-producing CD4+ T cells compared to suppressed PLWH (p=0.002) and HIV-seronegative individuals (p=0.0006) (*Figure 1B*). There was no significant difference in SARS-CoV-2-specific IFN-γ/TNF-α-producing CD8+ T cells among the groups (*Figure 1B*), and no significant differences in SARS-CoV-2-specific CD4+ or CD8+ T cell frequencies were observed between the suppressed PLWH and HIV-seronegative individuals (*Figure 1B*).

Simultaneous production of cytokines, commonly referred to as polyfunctionality, which is regarded as a measure of the quality of the T cell response, has been shown to correlate with viral control (*Betts et al., 2006*). Thus, we evaluated the quality of the CD4+ and CD8+ T cell responses among the groups by enumerating cells producing three (IFN-γ, TNF-α, and IL-2) cytokines in various

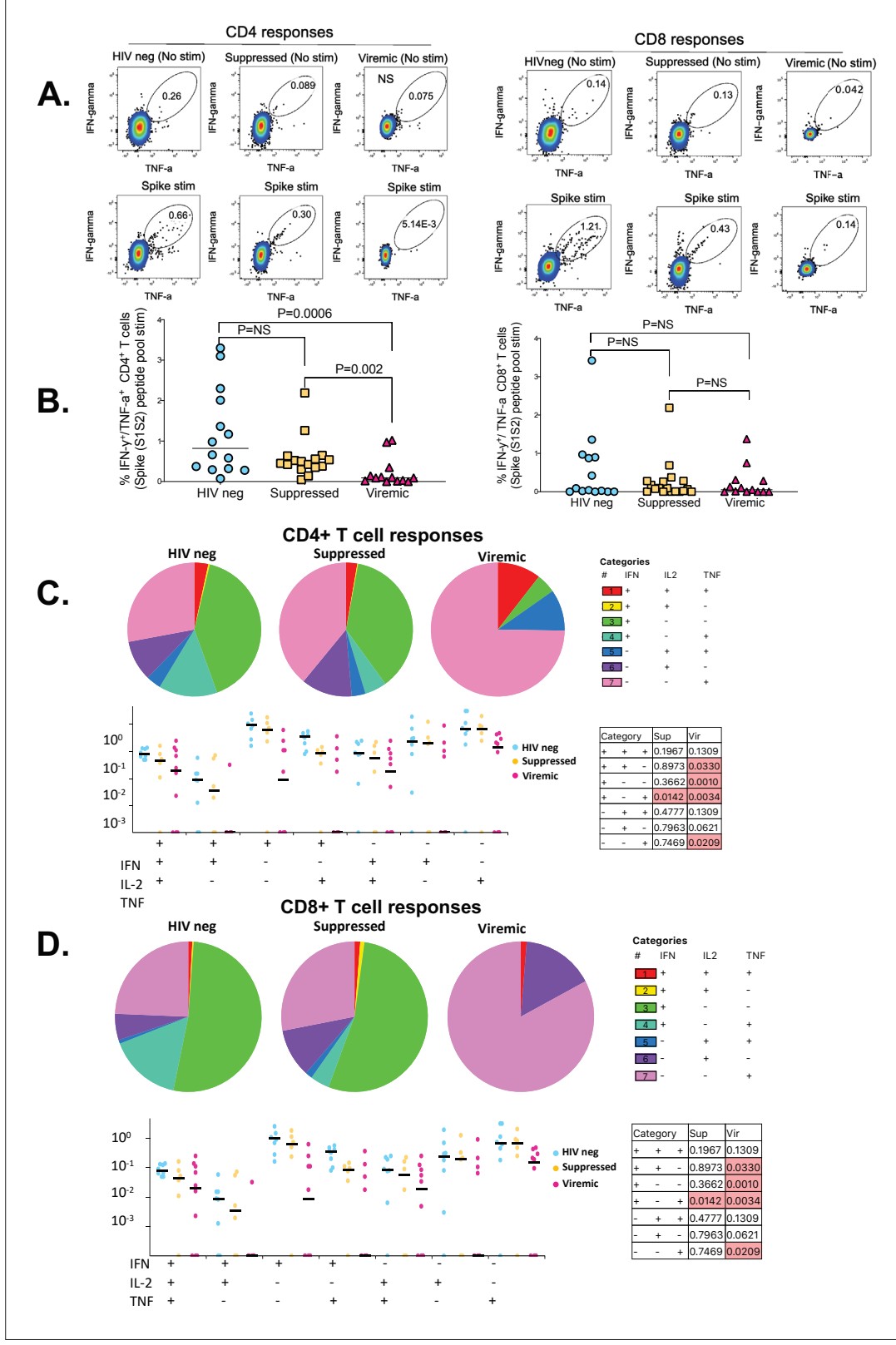

**Figure 1.** The impact of unsuppressed HIV infection on SARS-CoV-2-specific CD4+ and CD8+ T cell responses. (**A**) Representative flowplots gated on IFN-γ/TNF-α dual positive CD4+ and CD8+ T cells. (**B**) Aggregate data for IFN-γ/TNF-α dual positive CD4+ and CD8+ T cells are shown (HIV-neg, n=14; suppressed, n=16: viremic, n=13). SARS-CoV-2-specific CD4+ and CD8+ T cells producing IFN-γ, TNF-α, and IL-2 cells in various combinations are

*Figure 1 continued on next page*

*Figure 1 continued*

shown. Pie chart and dot plots for (**C**) SARS-CoV-2-specific CD4+ and (**D**) CD8+ T cells. Pie chart represents the mean distribution across subjects of mono-functional, bi-functional, and poly-functional cytokine producing SARS-CoV-2-specific T cells. Size of each pie segment relates to the frequency of a mono-functional, bi-functional, and triple-functional response. Dot plot represents the frequency of combinations of cytokines produced. Wilcoxon test was done among the dot plots using SPICE software (significant p values are highlighted).

combinations. Consistent with dual IFN-γ, TNF-α cytokine secretion data (*Figure 1B*), the patterns of cytokine production of HIV-seronegative was mostly similar to HIV suppressed individuals (pie charts, *Figure 1C and D*). Analysis of single cytokine production revealed that HIV-seronegative individuals and suppressed PLWH predominantly produced IFN-γ responses (green sectors of the pie chart, *Figure 1C and D*), whereas viremic PLWH predominantly produced TNF-α responses for both CD4+ and CD8+ T cells (magenta sectors of the pie chart, *Figure 1C and D*). Cells co-producing all three cytokines were very rare regardless of HIV status (red sectors of the pie chart, *Figure 1C and D*). Nonetheless, HIV-seronegative had greater frequencies of dual cytokine secreting cells compared to viremic PLWH (p=0.0330 for CD4, *Figure 1C*; p=0.0330 for CD8, *Figure 1D*). Taken together, the data show that uncontrolled HIV infection lowers the magnitude and alters the quality of SARS-CoV-2 T cell responses. Importantly, complete plasma HIV suppression preserves the capacity to mount high magnitude, dual-functional SARS-CoV-2-specific T cell responses.

## T cell responses against the major SARS-CoV-2 structural proteins

Having observed differences in magnitude and quality of SARS-CoV-2 spike-specific T responses, we next measured responses directed against major structural proteins, the nucleocapsid (N), the membrane (M), and Spike (S), again using PepTivater peptide pools from Miltenyi biotec. Our data show all three major SARS-CoV-2 proteins are targeted by SARS-CoV-2-specific CD4+ and CD8+ T cells (*Figure 2A and B*), with a preponderance for greater S-specific CD8+ T cell responses relative to M (*Figure 2A*). These data suggest that most SARS-CoV-2 structural proteins can be targeted by T cells, consistent with previous reports (*Tarke et al., 2021*).

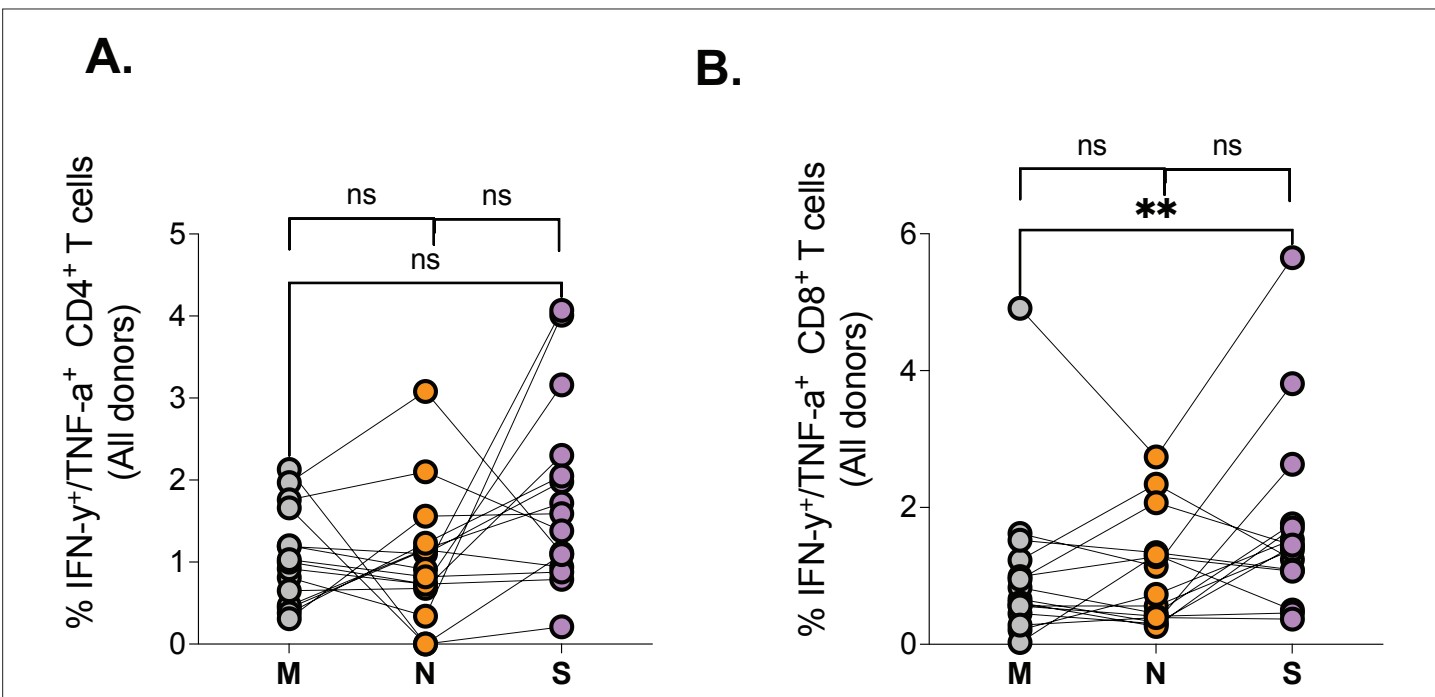

**Figure 2.** Comparison of SARS-CoV-2 protein targeting by T cell responses among HIV-negatives, suppressed and viremic donors. Magnitude of (**A**) CD4+ T and (**B**) CD8+ T cell responses targeting the Membrane (M), Nucleocapsid (N), and Spike (S) SARS-CoV-2 proteins among study groups. P values for differences among the groups are *<0.05; as determined by the Wilcoxon matched-pairs signed rank test (GraphPad Prism version 9.3.0).

## Uncontrolled HIV infection abrogates SARS-CoV-2 T cell cross-recognition between wild-type D614G and Beta variant

To evaluate the impact of uncontrolled HIV infection on cross-reactive T cell responses between wt and the Beta variant, we compared the breadth of responses and the ability to cross-recognize SARS-CoV-2 Beta variant peptides among the three study groups. These studies were conducted using two sets of 15 mer overlapping peptides (OLPs). Set 1 was comprised of 16 wt peptides, spanning the receptor-binding domain (RBD) and non-RBD regions of spike (S) that are known hotspots for mutations (*Tegally et al., 2021c*). Set 2 consisted of corresponding peptides that included all the major mutations that define the Beta variant lineage (*Wibmer et al., 2021*). A detailed description of the peptides is contained in *Supplementary file 1*.

We first sought to determine cross-reactivity of SARS-CoV-2-specific CD4$^+$ and CD8$^+$ T cells induced following infection with the wt (D614G, wave 1) and Beta variant (wave 2), between each other. We found that wave 1 donors had significantly lower CD8$^+$ (p=0.0312) and CD4$^+$ T cell responses (p=0.0078) to Beta variant relative to corresponding wt responses (*Figure 3A*). Wave 2 donors had no significant differences in T cells responses to Beta and wt (*Figure 3B*). Using a 12-day cultured stimulation assay, we were able to massively expand the magnitude of SARS-CoV-2-specific CD4$^+$ and CD8$^+$ T cells (*Figure 3C*) and (*Figure 3—figure supplement 1*), and this allowed us to hone in on single peptide responses (*Table 1*). Representative data for a wave 1 donor shows three CD8$^+$ and two CD4$^+$ wt responses (red circles), that did not cross-recognize corresponding Beta variants (blue bars) (*Figure 3D*). Contrariwise, a representative wave 2 donor had one CD8$^+$ and one CD4$^+$ T cell response to the Beta variant that did not cross-react to the wt version of the peptide (*Figure 3E*). Intra-donor comparison revealed significantly more CD8$^+$ (p=0.0156) and CD4$^+$ T cell responses (p=0.0312) to wt peptides compared to the corresponding Beta variant peptides in wave 1 donors (*Figure 3F*). Conversely, unlike the ex vivo data (*Figure 3B*), wave 2 donors had significantly more CD8$^+$ T cell responses to Beta variant peptides relative to wt peptides (p=0.0312), and a trend toward increased CD4$^+$ T cells against Beta peptides (p=0.0625), highlighting the increased sensitivity of expanded cells (*Figure 3G*). Taken together, these data show poor cross-recognition of wt and Beta variant epitopes.

We then assessed the impact of HIV infection on cross-recognition of wt and Beta variant epitopes. Representative data for an HIV-seronegative individual from the first wave had six wt and five Beta variant CD8$^+$ T cell responses, one was cross-recognized (circled) (*Figure 4A*). The same individual had five wt and five Beta variant CD4$^+$ T responses, one was cross-recognized (*Figure 4B*). Similarly, a representative suppressed wave 1 donor had five wt and two Beta variant CD8$^+$ T cell responses, one of which was cross-recognized (*Figure 4C*). This same donor had six wt and zero Beta variant CD4$^+$ T cell responses (*Figure 4D*). A representative viremic individual had four weak wt CD8$^+$ T cell responses and three borderline CD4 responses, none of which were cross-recognized (*Figure 4E and F*). Summary data showed viremic PLWH had significantly narrow breadth of SARS-CoV-2-specific CD8$^+$ (p=0.039) and CD4$^+$ T cell responses (p=0.033) compared to suppressed PLWH and HIV-seronegative individuals (*Figure 4G and H*). Collectively, these data show that SARS-CoV-2-specific T cell responses in viremic PLWH have limited breadth and subsequently poor cross-recognition potential.

## Identification of mutations in the Beta variant that are associated with reduced cross-recognition

Having shown poor T cell cross-recognition of SARS-CoV-2 epitopes between wt and Beta variant, we next sought to identify mutations that might be responsible for the loss of recognition. We combined all the T cell data for the 12 (4 HIV-negatives, 4 HIV-suppressed, and 4 HIV-viremics) donors used for cultured epitope screening studies. All the samples were culturally expanded using wt peptides from the first wave. This analysis identified four Beta variant peptides (listed in *Supplementary file 1*) that had significant reduction in CD8$^+$ T cell recognition relative to wt peptides (*Figure 5A*). Three of these peptides were also poorly recognized by CD4$^+$ T cells (*Figure 5B*). The amino acid sequences for wt and corresponding mutations include the E484K mutation, a key Beta variant spike residual change also associated with loss antibody binding (*Wibmer et al., 2021*). Taken together, these data identified mutations in the Beta variant that may abrogate T cell recognition, suggesting that they may be potential T cell escape mutations and warrant further investigation.

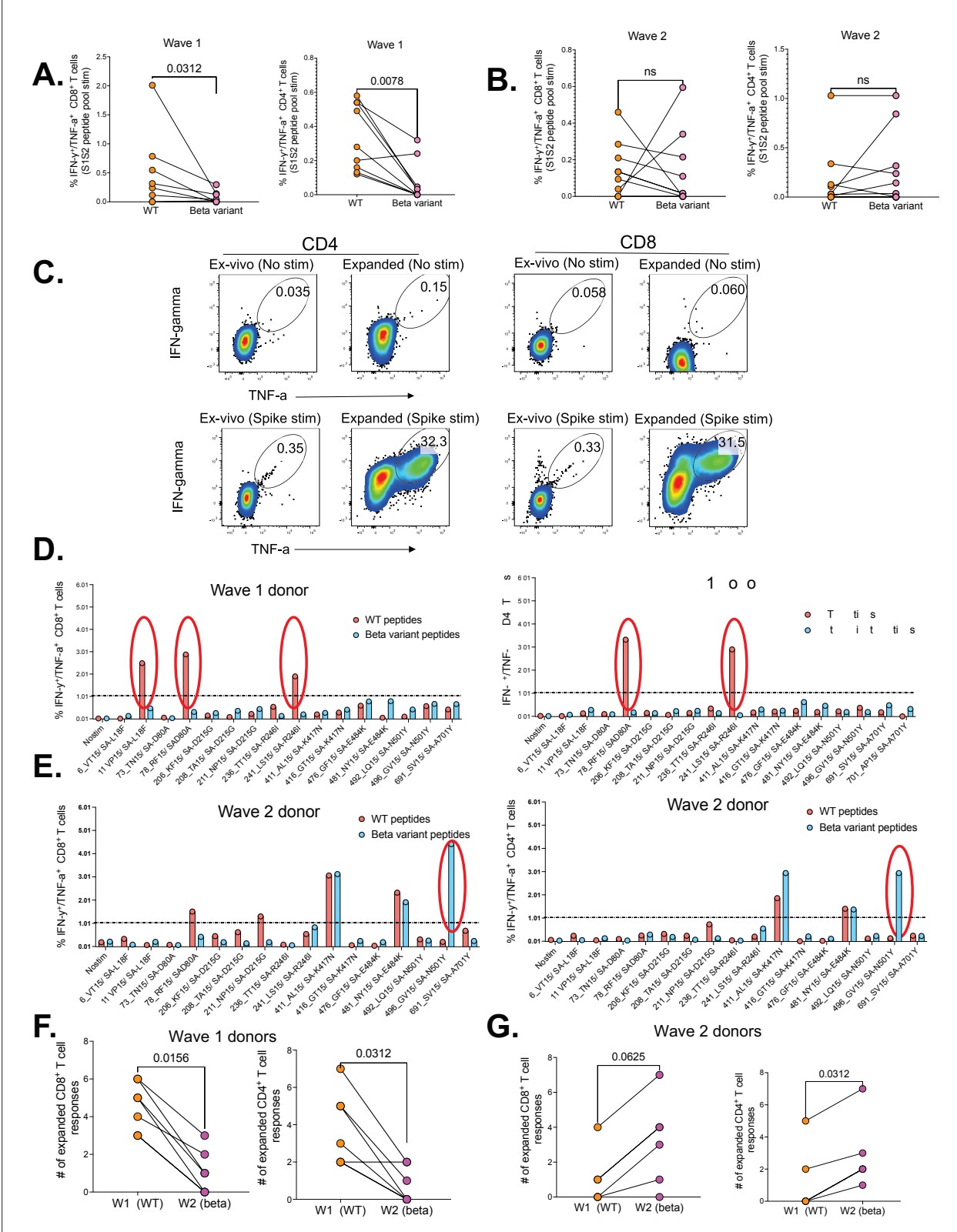

**Figure 3.** Poor cross-recognition of SARS-CoV-2-specific CD4[+] and CD8[+] T cell responses between wt and beta variants in wave 1 and wave 2 COVID-19 participants. Ex vivo assessment of T cell cross-recognition between the two waves. (**A**) Intra-donor SARS-CoV-2-specific T cell responses to wt and corresponding Beta variant peptides by wave 1 participants. (**B**) Intra-donor SARS-CoV-2-specific T cell responses to wt and corresponding Beta variant peptides in wave 2 participants. Next, PBMCs were expanded for 12 days in the presence of S1S2 SARS-CoV-2 peptide pools and tested against wt

*Figure 3 continued on next page*

*Figure 3 continued*

and corresponding Beta variants at single peptide level. (**C**) Representative flow plots showing the frequency of SARS-CoV-2-specific CD4+ and CD8+ T cells before and after cultured expansion. (**D**) T cell responses to single wt (red bars) and corresponding Beta (blue bars) peptide stimulation for a representative donor from wave 1. (**E**) T cell responses to single wt and corresponding Beta peptide stimulation for a representative donor from wave 2 (positive responses are circled). A response was deemed positive if ≥1% or higher. (**F**) Number of expanded wt and corresponding Beta responses for each wave 1 donor. (**G**) Number of expanded wt and corresponding Beta responses for each wave 2 donor. P values calculated using Wilcoxin matched-pairs signed rank T test. PBMC, peripheral blood mononuclear cell; wt, wild-type.

The online version of this article includes the following figure supplement(s) for figure 3:

**Figure supplement 1.** Cross-recognition of SARS-CoV-2 CD4+ T cell responses between wt and Beta variants in wave 1 and wave 2 COVID-19 donors: PBMCs were expanded for 12 days in the presence of S1S2 SARS-CoV-2 peptide pools.

## Immunodominance hierarchy of SARS-CoV-2 CD8+ and CD4+ T cell responses targeting the spike protein

Virus-specific CD8+ and CD4+ T cells typically target viral epitopes in a distinct hierarchical order (*Streeck et al., 2009*; *Laher et al., 2017*). Identifying SARS-CoV-2 epitopes that are most frequently targeted by T cells is important for the design of vaccines that can induce protective T cell responses. To determine the immunodominance hierarchy of SAR-CoV-2 specific T cell responses targeting the spike protein, OLPs were ranked based on magnitude and frequency of recognition. This analysis revealed the most immunodominant wt peptides targeted by CD8+ T cell responses (*Figure 6A*). The Beta variant resulted in dramatic shift in the immunodominance hierarchy whereby, three of five most dominant wt CD8+ T cell responses (*Figure 6A*), their Beta variant versions were subdominant (downward arrows) (*Figure 6B*). Contrariwise, three subdominant wt responses were among the most dominant Beta variant responses (upward arrows) (*Figure 6B*). A similar trend was observed for CD4+ T cell responses (*Figure 6C and D*). These data demonstrated a shift in the immunodominant hierarchy between wt and Beta variant responses, which partly explains poor T cell cross-recognition between successive SARS-CoV-2 variants.

## The impact of HIV markers of diseases progression on SARS-CoV-2-specific T cell responses

To gain more insight into why viremic PLWH responded poorly to SARS-CoV-2 infection, we investigated if T cell activation defined here as co-expression of CD38 and HLA-DR, absolute CD4 count and plasma viral load, impacted immune responses (*Du et al., 2009*). The proportion of activated (CD38/HLA-DR) CD4+ T cells was higher in viremic PLWH compared to suppressed (p=0.02) and HIV-seronegative individuals (p=0.002; *Figure 7A*). Moreover, proportion of activated (CD38/HLA-DR) CD4+ T cells among viremic PLWH negatively correlated with absolute CD4 counts (r=–0.7, p=0.04; *Figure 7B*), and positively correlated with HIV plasma viral loads (r=0.9, p=0.0004; *Figure 7C*). Similarly, proportion of activated (CD38/HLA-DR) CD8+ T cells were significantly higher in viremic PLWH relative to suppressed PLWH (p=0.04) and HIV-seronegative individuals (p=0.0008; *Figure 7D*). The negative relationship between proportion of activated (CD38/HLA-DR) CD8+ T cells and CD4 counts did not reach statistical significance (*Figure 7E*), but proportion of activated (CD38/HLA-DR) CD8+ T cells were positively correlated with HIV plasma viral loads among viremic PLWH (r=0.8, p=0.0006; *Figure 7F*).

Taken together, these data suggest that hyper immune activation driven by uncontrolled HIV infection impacts CD4+ and CD8+ T cell responses.

Finally, we interrogated the relationship between SARS-CoV-2-specific responses and disease severity, stratified into asymptomatic, mild, and severe diseases requiring oxygen supplementation, as previously defined (*Karim et al., 2021a*). We found no significant differences between the magnitude of CD4+ or CD8+ T cell responses and disease severity among the groups (*Figure 7—figure supplement 1A,B*). We next, examined sex differences and found no difference in CD4+ and CD8+T cell responses to SARS-CoV-2 infection (*Figure 7—figure supplement 1C,D*). Age is a risk factor for severe COVID-19 (*WHO, 2021a*); thus, we examined the relationship between age and T cell responses. There was a negative relationship between age and magnitude of CD8+ T cell responses (CD8 r=−0.6, p=0.002) (*Figure 7—figure supplement 1E*), and a similar trend for CD4+ T cell responses (CD4 r=−0.3, *P*=0.15) (*Figure 7—figure supplement 1F*). These data show that younger

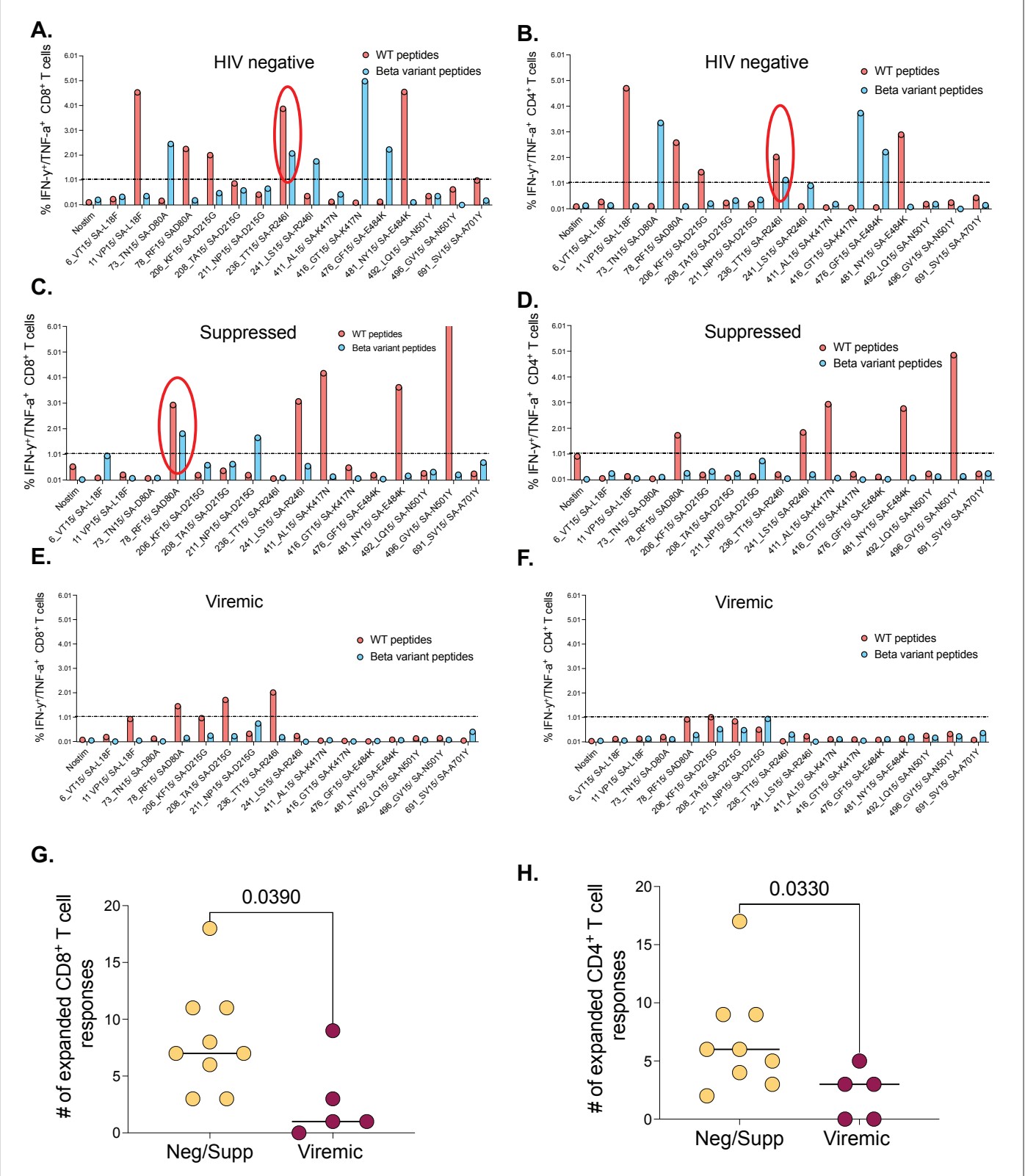

**Figure 4.** The effects of unsuppressed HIV infection on T cell breadth and ability to cross-recognize the Beta variant. Representative data for a negative donor showing greater, (**A**) CD8[+] and (**B**) CD4[+] T cell breadth. A cross-recognized responses between wt and Beta is circled. Representative data for a suppressed donor showing greater, (**C**) CD8[+] and (**D**) CD4[+] T cell breadth. A cross-recognized response is circled. Representative data for a viremic donor showing greater, (**E**) CD8[+] and (**F**) CD4[+] T cell breadth. (**G**) Aggregate data comparing breath of SARS-CoV-2-specific CD8[+], and (**H**) CD4[+] T

*Figure 4 continued on next page*

*Figure 4 continued*

cell response between HIV-negative and suppressed versus viremics. Breadth here is simply the number of positive responses among the individual peptides tested.

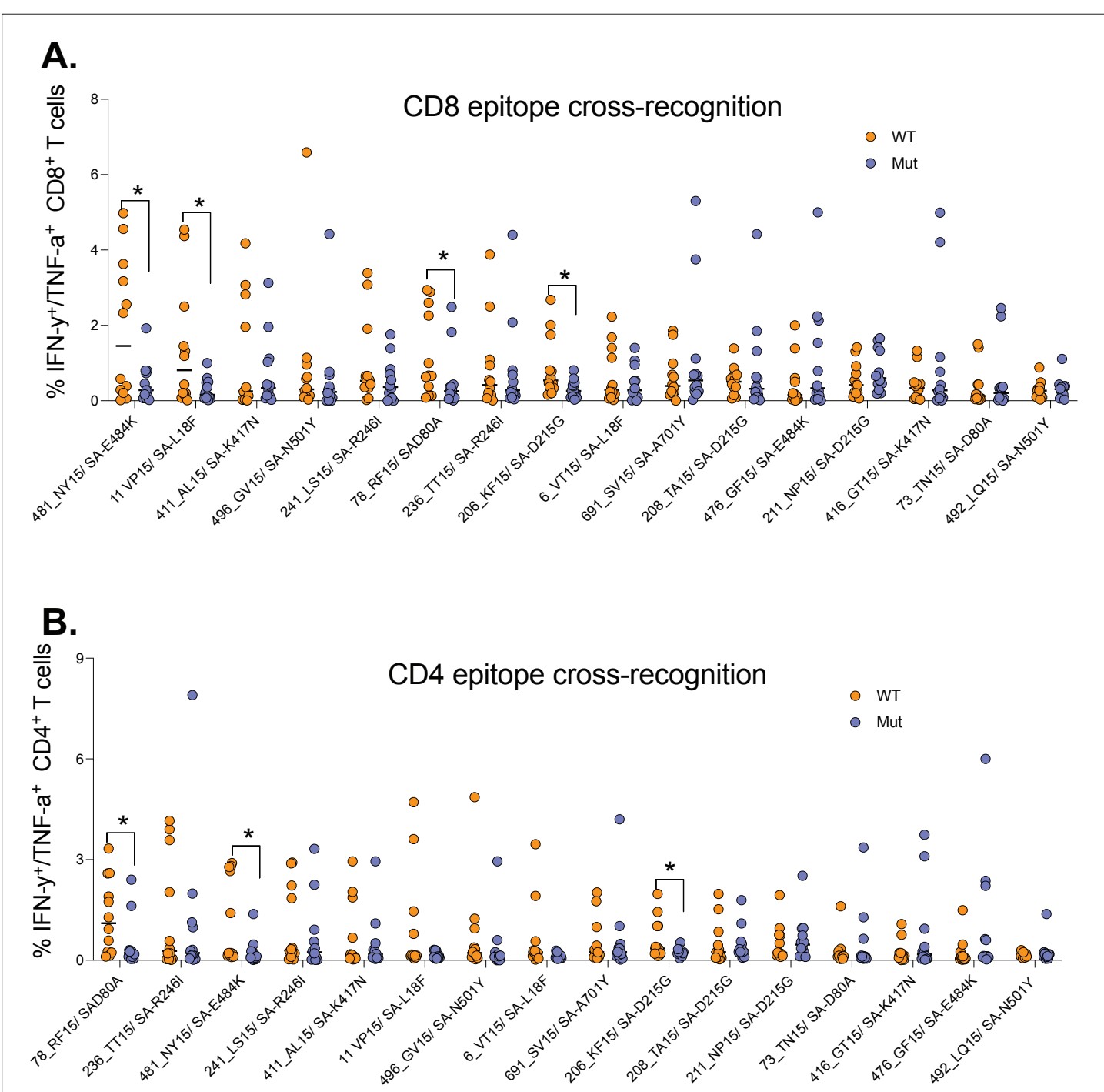

**Figure 5.** Identification of Beta mutations associated with reduced cross-recognition between wt and Beta variant. (**A**) Side-by-side comparison of SARS-CoV-2-specific CD8+ T cell response between wt and Beta. (**B**) Side-by-side comparison of SARS-CoV-2-specific CD4+ T cell response between wt. The analysis combined all the 12 participants. P values calculated by Mann-Whitney U-test. wt, wild-type.

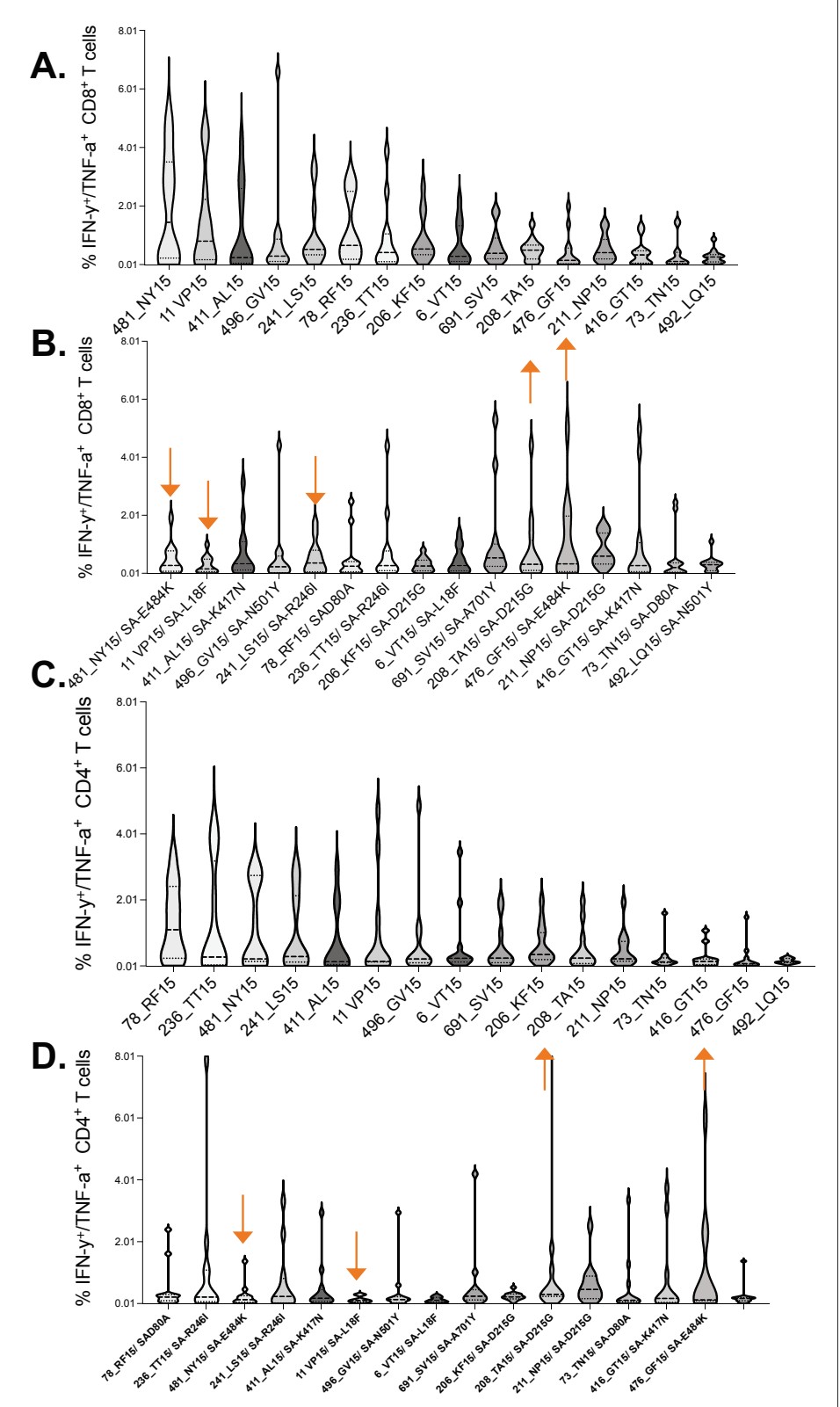

**Figure 6.** Immunodominance hierarchy of SARS-CoV-2 CD8[+] and CD4[+] T cell responses targeting wt and Beta. Immunodominance hierarchy of CD8[+] T cell responses to, (**A**) wt and (**B**) the corresponding Beta variant peptides. Similarly, Immunodominance hierarchy of CD4[+] T cell responses to, (**C**) wt and (**D**) the corresponding Beta variant. Arrows indicate responses that changed hierarchical position (among the six most dominant responses) between

*Figure 6 continued on next page*

*Figure 6 continued*

the two waves. Data arranged in descending order of magnitude of responses to wt peptide stimulation. wt, wild-type.

people had greater responses compared to older people, whereas disease severity and sex did not have discernible effect on SARS-CoV-2 T cell responses.

## Discussion

The greater burden of HIV in sub-Saharan Africa makes investigating the impact of HIV infection on COVID-19 immunity and disease outcomes critical for bringing the epidemic under control in the region. Recent studies have documented strong cellular responses following SARS-CoV-2 infection and vaccination, but the effects of HIV on SARS-CoV-2-specific T cell responses are not well characterized. Here, we investigated the antigen-specific CD4$^+$ and CD8$^+$ T cell responses in a cohort of SARS-CoV-2- infected individuals with and without HIV infection. Our results show that unsuppressed HIV infection is associated with reduced cellular responses to SARS-CoV-2 infection. We also show that low absolute CD4 count and hyper immune activation are associated with diminution of SARS-CoV-2-specific T cell responses. Importantly, we identify spike mutations in the Beta variant that abrogate recognition by memory T cells raised against wt epitopes. Similarly, immune responses targeting Beta variant epitopes poorly cross-recognize corresponding wt epitopes. These data reveal the potential

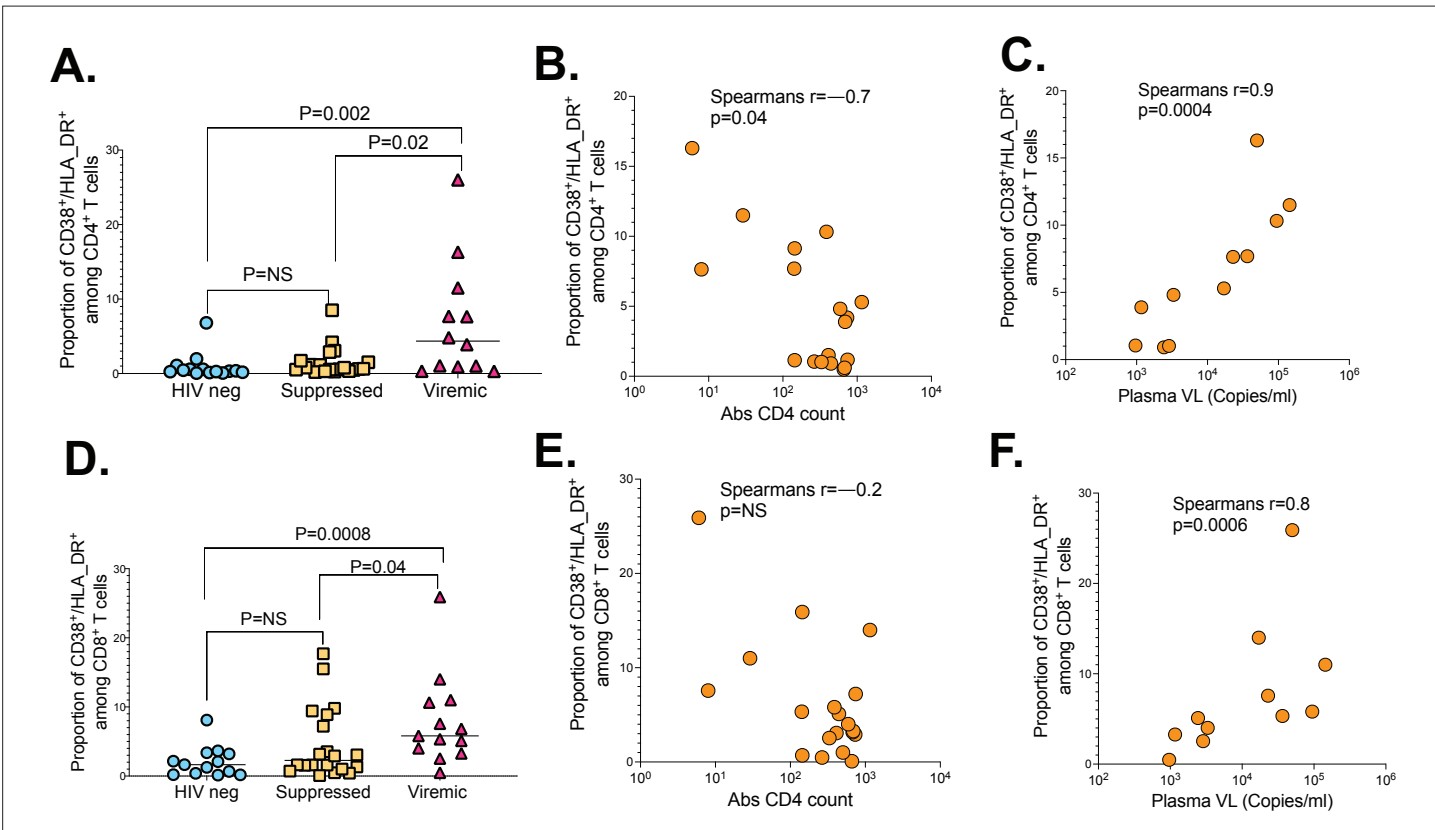

**Figure 7.** The impact of HIV markers of diseases progression on SARS-CoV-2 T cell immunity. (**A**) CD4$^+$ T cell activation graphed based on the frequency of CD38/HLA-DR co-expressing cells. (**B**) Correlation between CD4$^+$ T cell activation and absolute CD4 counts of viremic PLWH. (**C**) Correlation between CD4$^+$ T cell activation and HIV plasma viral load of viremic PLWH. (**D**) CD8$^+$ T cell activation measured by CD38/HLA-DR. (**E**) Correlation between CD8$^+$ T cell activation and absolute CD4 counts of viremic PLWH. (**F**) Correlation between CD8$^+$ T cell activation and HIV plasma viral load of viremic PLWH. P values calculated by Mann-Whitney U-test and Pearson correlation test. PLWH, people living with HIV.

The online version of this article includes the following figure supplement(s) for figure 7:

**Figure supplement 1.** Assessment of the effect of COVID-19 disease severity on, (**A**) SARS-CoV-2-specific CD4$^+$, and (**B**) CD8$^+$ T cell responses.

for emerging SARS-CoV-2 variants to escape T cell recognition. Importantly, our data highlight the potential for unsuppressed HIV infection to attenuate vaccine-induced T cell immunity.

HIV-induced immune dysregulation is well documented (*Klatt et al., 2013*). Unsuppressed HIV infection is associated with profound dysfunction of virus-specific T cell immunity partly caused by immune activation (*Klatt et al., 2013*; *Ndhlovu et al., 2015*). Recent studies have reported strong association between unsuppressed HIV infection and poor COVID disease outcomes, for instance, a large cross-section study found a link between severe HIV disease and poor COVID-19 outcomes including COVID-19-associated death (*Chanda et al., 2020*). This study showed that individuals with unsuppressed HIV infection mount weak responses to SARS-CoV-2 infection and poorly recognize SARS-CoV-2 Beta variant mutations. We also examined several mechanisms by which unsuppressed HIV can impact SARS-CoV-2-specific T cell responses and found that HIV-induced immune defects such as low CD4$^+$ T cell counts, higher HIV plasma viral loads, and elevated immune activation were invariably associated with diminished SARS-CoV-2 responses. These findings are consistent with several recent reports, such as a case of one HIV-positive patient with low CD4 count that had prolonged CIVID-19 disease (*Wang et al., 2020*). The ability of unsuppressed HIV to cause severe immune activation was also recently documented by others (*d'Ettorre et al., 2020*; *Sharov, 2021*). Taken together, these data suggest that HIV-induced immune dysregulation negatively impacts the potential to mount robust T cell responses to SARS-CoV-2 infection.

Furthermore, although ART-mediated HIV suppression rarely results in complete immune reconstitution (*Henrich et al., 2017*), sustained complete plasma HIV suppression was associated with robust SARS-CoV-2 responses that were mostly similar in magnitude and quality to responses mounted by HIV-seronegative individuals. Given reduced levels of CD38 and HLA-DR dual positive cells and near normal absolute CD4 counts in suppressed individuals, it is reasonable to speculate that reduced immune activation and superior CD4$^+$ T helper function were partly responsible for improved immune responses in suppressed individuals.

The emergence of several SARS-CoV-2 variants with mutations in the viral Spike (S) protein such as mutations in the RBD, N-terminal domain (NTD), and furin cleavage site region (*Tarke, 2021*) continue to fuel the epidemic. These mutations have been shown to directly affect ACE2 receptor binding affinity, infectivity, viral load, and transmissibility (*Tarke, 2021*; *Greaney et al., 2021*; *Starr et al., 2021*). The variants of concern identified since the start of the COVID-19 pandemic include the Alpha (*Davies, 2020b*), Beta (*Tegally et al., 2021b*), Gamma (*Voloch et al., 2020*), and Delta (*Mallapaty, 2021*), and now the Omicron variant. Most of these have been shown to attenuate neutralization but the impact of these mutations on T cell responses has not been extensively explored (*Riou et al., 2022*). However, a recent report demonstrating the potential for SARS-CoV-2 to evade cytolytic T lymphocyte (CTL) surveillance, highlight the need for more investigations regarding the potential CTL-driven immune pressure to shape emerging variants (*Agerer et al., 2021*). To this end, our study provides new evidence that SARS-CoV-2 has the potential to evade T cell recognition. Moreover, our data suggest that spike mutations in the Beta variant that were associated with antibody escape may also escape T cell recognition.

Southern Africa has had at least four epidemic waves of COVID-19. The first was a mixture of SARS-CoV-2 lineages (with D614G), the second wave was driven by the Beta variant (*Tegally et al., 2021a*), and the third by the Delta variant (*Callaway, 2021*). The fourth wave dominated by the highly mutated Omicron variant (*WHO, 2021b*; *Viana et al., 2021*). Intriguingly, there was some evidence that PLWH in South Africa had increased disease severity in the second wave compared to the first wave (*Karim et al., 2021a*). The precise mechanisms responsible for increased severity are not fully understood, but low CD4$^+$ T cell counts and high neutrophil-to-lymphocyte ratio (NLR) showed strong association with disease severity (*Karim et al., 2021a*). Our data suggest that diminished T cell responses to the Beta variant even in previously exposed individuals may have contributed to severe disease in the second wave.

Here, we report poor cross-recognition of the Beta variant by individuals infected with wt and vice versa, which was exacerbated by unsuppressed HIV infection. However, others have reported better cross-recognition between variants and vaccines. Possible explanation for the apparent discrepancy include, (1) unlike other studies that compared responses to the entire spike protein using peptide pools to stimulate cells (*Keeton et al., 2022*; *Gao et al., 2022*), our cross-recognition studies focused on head-to-head comparisons of single wt peptides with corresponding variants peptides

containing a lineage defining mutation (*Keeton et al., 2022*). We may have picked up fewer cross-reactive responses because we used dual section of IFN-γ and TNF-α as a readout for antigen-specific responses, which is more stringent than single cytokine producing cells. (3) We used cultured expansions prior to ICS assays which amplifies the response several folds above background and therefore more specific. In fact, our ex vivo cross-recognition data are comparable to other studies which also showed diminution of responses across variants (*Keeton et al., 2022*). Future studies should apply our cultured expansion and the dual cytokine secretion readout to assess cross-recognition among other variants and different vaccine regimens.

Although, we repeatedly showed robust in vitro T cell expansion following ex vivo peptide stimulation but limited expansion against mutant versions of the peptides, there is need to identify optimal peptides that were targeted by CD8+ and CD4+ T cells in the context of restricting MHC class I and II alleles. SARS-CoV-2 responses are generally very broad (*Grifoni et al., 2020*); thus, it is not clear from these studies how the loss of T cell cross-recognition in Spike affects the overall protective immunity. Furthermore, investigating if the observed poor T cell cross-recognition between wave 1 and wave 2 is generalizable to the Delta and the Omicron variants is clearly warranted. Importantly, our data raise the question of whether CTL selection pressure plays a significant role in shaping emerging variants. This concept should be investigated using larger longitudinal studies with longer durations of follow-up.

Previous work in this cohort examined the relationship T cell and B cell responses and found a positive association between CD8+ T cells frequency and several CD19 B cell subsets, which was attenuated in PLWH (*Karim et al., 2021a*), suggesting that both arms of the immune system are impacted by HIV/SARS-CoV-2 coinfection. However, the current study did not examine this relationship at antigen-specific level due to sample limitations. Future work is required to understand the relationship between T cell and humoral immunity and the impact of unsuppressed HIV infection on long-term protection.

In conclusion, we show that uncontrolled HIV infection is associated with low magnitude, reduced polyfunctionality, and diminished cross-recognition of SARS-CoV-2-specific CD4+ and CD8+ T cell responses. Importantly, fully suppressed PLWH had comparable SARS-CoV-2-specific T cell responses with HIV-seronegative individuals. These findings may partly explain high propensity for severe COVID-19 among PLWH and also highlight their vulnerability to emerging SARS-CoV-2 variants of concern, especially those with uncontrolled HIV infection. Hence, there is need to ensure uninterrupted access to ART for PLWH during the COVID-19 pandemic.

## Materials and methods

### Ethical declaration

The study protocol was approved by the University of KwaZulu-Natal Biomedical Research Ethics Committee (BREC) (approval BREC/00001275/2020). Consenting adult patients (>18 years old) presenting at King Edward VIII, Inkosi Albert Luthuli Central Hospital, and Clairwood Hospital in Durban, South Africa, between July 29 and August November 2021 with PCR confirmed SARS-CoV-2 infection were enrolled in the study.

### Sample collection and laboratory testing

Blood samples used in this study were collected between 1 and 3 weeks after COVID-19 PCR positive diagnosis. HIV testing was done using a rapid test and viral load quantification was performed from a 4-ml EDTA by a commercial lab (Molecular Diagnostic Services, Durban, South Africa) using the Real-Time HIV-negative1 viral load test on an Abbott machine. CD4 counts were performed by a commercial lab (Ampath, Durban, South Africa). PLWHs were categorized into suppressed and unsuppressed based on viral load measurements of <50 and >1000 copies/ml, respectively, at the time of sample collection.

### T lymphocyte phenotyping

PBMCs were isolated from blood samples by density gradient method and cryopreserved in liquid nitrogen as previously described (*Karim et al., 2021b*). Frozen PBMCs were thawed, rested, and stimulated for 14 hr at 37°C, 5% $CO_2$ with either staphylococcal enterotoxin B (SEB, 0.5 µg/ml), SARS-CoV-2

wt peptide pool (8 µg/ml), 501Y.V2 variant peptide pool (4 µg/ml), or the Control Spike peptide pool (Miltenyi, Bergisch Gladbach, Germany, 2 µg/ml). Brefeldin A (BioLegend, CA) and CD28/CD49d (BD Biosciences, Franklin Lakes, NJ) were also added ahead of the 14-hr incubation at 5 and 1 µg, respectively. The cells were stained with an antibody cocktail containing: Live/Dead fixable aqua dead cell stain, anti-CD3 PE-CF594 (BD), anti-CD4 Brilliant Violet (BV) 650, anti-CD8 BV 786 (BD), anti-CD38 Alexa Fluor (AF) 700 (BD), anti-human leukocyte antigen (HLA) – DR Allophycocyanin (APC) Cy 7 (BD), and anti-programmed cell death protein 1 (PD) BV 421 (BD). After a 20-min incubation at room temperature, the cells were washed, fixed, and permeabilized using the BD Cytofix/Cytoperm fixation permeabilization kit. Thereafter, the cells were stained for 40 min at room temperature with an intracellular antibody cocktail containing: anti-IFN-γ BV 711 (BD), anti-IL-2 PE (BD), and anti-TNF-α PE-Cy 7 (BD). Finally, the cells were washed and acquired on an LSR Fortessa and analysed on FlowJo v10.7.2. Differences between groups were considered to be significant at a p value of <0.05. Statistical analyses were performed using GraphPad Prism 8.0 (GraphPad Software, Inc, San Diego, CA).

## Ex-vivo cultured expansion of SARS-COV-2-specific T cells

PBMCs at a concentration of 2 million cells per well in a 24-well plate in R10 medium were stimulated with 10 µg/ml of SARS-COV-2 of OLP pools spanning the entire spike protein. The cells were incubated at 37°C in 5% $CO_2$. After 2 days, the cells were washed and fresh R10 medium supplemented with 100 U/ml recombinant IL-2 was added. Cultured cells were fed twice weekly with regular medium replenishment. On day 14, the cells were washed three times with fresh R10 medium and rested at 37°C in 5% $CO_2$ overnight in fresh R10 medium. On the following day, the cells were restimulated with individual peptides for 16 hr followed by ICS. Peptides that induced IFN-γ/TNF-α dual production above background (No stimulation control) were deemed reactive. Meaning that the expanded cells contained a subset of cells that were specific for that particular peptide.

## Statistical analyses

All statistical analyses were conducted with GraphPad Prism 9.3.1 (GraphPad Software, La Jolla, CA) and p values were considered significant if less than 0.05. Specifically, the Mann-Whitney U- and Kruskal-Wallis H-tests were used for group comparisons. Additional post hoc analyses were performed using the Dunn's multiple comparisons test. Correlations between variables were defined by the Spearman's rank correlation test. Categorical data were analysed using the Fisher's exact test.

## Acknowledgements

The authors would like to thank our study participants, the laboratory and clinic staff at Africa Health Research Institute for collecting the samples and compiling the clinical demographic data for the study. The authors would like to thank Drs Wendy Burgers, Catherine Riou, and Robert Wilkinson for designing and providing us the SARS-CoV-2 wt and Beta variant peptides. The authors thank Ms Anele Mbata and Mr Mza Nsimbi for their assistance with sample processing and cryopreservation.

## Additional information

### Competing interests
COMMIT-KZN-Team: The other authors declare that no competing interests exist.

### Funding

| Funder | Grant reference number | Author |
| --- | --- | --- |
| Howard Hughes Medical Institute | 55008743 | Zaza M Ndhlovu |
| Bill and Melinda Gates Foundation | INV-018944 | Alex Sigal |
| South Africa Medical Research Council | 31026 | Willem Hanekom |

| Funder | Grant reference number | Author |
| --- | --- | --- |
| Sub-Sahara African Network for TB and HIV Research Excellence | COL016 | Zaza M Ndhlovu |
| Africa Health Research Institute | LoA R82 | Zaza M Ndhlovu |

The funders had no role in study design, data collection and interpretation, or the decision to submit the work for publication.

## Author contributions

Thandeka Nkosi, Investigation, Writing - original draft, Writing – review and editing; Caroline Chasara, Investigation, Writing - original draft; Andrea O Papadopoulos, Resources, Formal analysis, Investigation, Writing - original draft, Project administration, Writing – review and editing; Tiza L Nguni, Investigation, Methodology, Writing - original draft; Farina Karim, Inbal Gazy, Data curation, Project administration; Mahomed-Yunus S Moosa, Resources, Investigation, Project administration, Writing – review and editing; Kondwani Jambo, Formal analysis, Writing – review and editing; COMMIT-KZN-Team, Data curation, Investigation, Project administration; Willem Hanekom, Conceptualization, Funding acquisition, Writing – review and editing; Alex Sigal, Data curation, Project administration, Writing – review and editing; Zaza M Ndhlovu, Conceptualization, Formal analysis, Validation, Investigation, Writing - original draft, Project administration, Writing – review and editing

## Author ORCIDs

Caroline Chasara http://orcid.org/0000-0001-6860-6111
Andrea O Papadopoulos http://orcid.org/0000-0001-5317-1418
Farina Karim http://orcid.org/0000-0001-9698-016X
Mahomed-Yunus S Moosa http://orcid.org/0000-0001-6191-4023
Alex Sigal http://orcid.org/0000-0001-8571-2004
Zaza M Ndhlovu http://orcid.org/0000-0002-2708-3315

## Ethics

Human subjects: Ethical Declaration: The study protocol was approved by the University of KwaZulu-Natal Biomedical Research Ethics Committee (BREC) (approval BREC/00001275/2020). Consenting adult patients (>18 years old) presenting at King Edward VIII, Inkosi Albert Luthuli Central Hospital, and Clairwood Hospital in Durban, South Africa, between 29 July to August November 2021 with PCR confirmed SARS-CoV-2 infection were enrolled into the study.

## Decision letter and Author response

Decision letter https://doi.org/10.7554/eLife.78374.sa1
Author response https://doi.org/10.7554/eLife.78374.sa2

# Additional files

## Supplementary files

• Supplementary file 1. Wild-type (wt) Spike overlapping peptides and corresponding Beta variant peptides. The table contains a list of peptides spanning the receptor-binding domain (RBD) and non-RBD regions of spike with known hotspots for mutations, and a corresponding list of peptides with Beta variant lineage defining mutations. The Beta variants mutations are highlighted in red. The two sets of peptides were used for cultured expansion studies.

• MDAR checklist

## Data availability

All source data files for the figures are now publicly available on our institutional website (Africa Health Research Institute database). The data can be accessed using this link: https://doi.org/10.23664/AHRI.SARS.CoV.2.

The following dataset was generated:

| Author(s) | Year | Dataset title | Dataset URL | Database and Identifier |
|---|---|---|---|---|
| Nkosi K, Charasa C, Papadopoulos AO, Nguni TZ, Karim F, Moosa MYS, Gazy I, Jambo K, Hanekom W, Sigal A, Ndhlovu ZM | 2022 | Unsuppressed HIV infection impairs T cell responses to SARS-CoV-2 infection and abrogates T cell cross-recognition | https://doi.org/10.23664/AHRI.SARS.CoV.2 | AHRI Data Repository, 10.23664/AHRI.SARS.CoV.2 |

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
