## [Editor Report]

This paper provides important descriptive evidence that untreated HIV infection has important negative effects on T cell responses to SARS-CoV-2, particularly in regards to cross recognition of new variants. Treatment of HIV with ART appears to partially reverse suppression of SARS-CoV-2 specific cellular immune responses.

---

## [Decision Letter]

**Decision letter after peer review:**

Thank you for submitting your article "Unsuppressed HIV infection impairs T cell responses to SARS-CoV-2 infection and abrogates T cell cross-recognition" for consideration by *eLife*. Your article has been reviewed by 3 peer reviewers, and the evaluation has been overseen by a Reviewing Editor and Tadatsugu Taniguchi as the Senior Editor. The following individual involved in the review of your submission has agreed to reveal their identity: Joshua T Schiffer (Reviewer #1).

Essential revisions:

1) Please demonstrate that the mean time between the PCR test and the analysis of the T cell responses is not a confounding variable that is biasing the result.

2) In figure 1, please check whether results are internally consistent across pie charts and histograms in panels C and D. If not, please provide an explanation.

3) References to other studies of CoV2 and HIV are incomplete and require greater attention in the intro and discussion. In particular, the cross-reactivity data seems to conflict with other studies showing cross-variant reactivity within a person. Please address this in the discussion.

4) Consider the addition of B cell and antibody data if this is available.

5) Please address reviewer 2's concerns about the compatibility of results in the text and figures 1 and 2.

*Reviewer #1 (Recommendations for the authors):*

In figure 1, the data does not seem internally consistent across pie charts and histograms in panels C and D. Please review and provide an explanation. References to other studies of CoV2 and HIV are incomplete and require greater attention in the introduction and discussion.

The cross-reactivity data seems to conflict with other studies showing cross-variant reactivity within a person. Please address this in the discussion.

*Reviewer #2 (Recommendations for the authors):*

With its limited impact on the field, it would have been a more complete manuscript if antibody responses against SARS-CoV-2 had been studied as well and linked with the T cell responses.

The first sentence of the abstract claims that HIV infection is one of the major risk factors for severe COVID-19. This statement is not true.

The sampling time points were from 2 to 4 weeks after the COVID-19 PCR test. The authors need to show the mean time between the PCR test and the analysis of the T cell responses to demonstrate that no difference if biasing the data. Indeed, T cell magnitude can vary between 2 and 4 weeks with earlier sampling showing higher T cell responses.

Figure 1B:

The authors claim in the text a significantly higher frequency of Spike-specific IFN-γ+ TNF-α+ CD8 T cells between healthy control and HIV viremic but the figure says NS.

Figures 1C and D:

The authors mentioned arcs around the pie chart representing specific cytokine contributions to the pies but there are no arcs in the figure. It would be better to show percentages of total IFN-γ+ cells and TNF-α+ cells within the responding cells.

Figure 2:

There was no significant difference in the percentage of Spike-specific IFN-γ+ TNF-α+ CD8 T cells between healthy control and HIV viremic in figure 1B. However, there is a significantly higher percentage of the cells in CD8 T cells in this figure. This is confusing. The data from only 4 viremic donors are not strong enough to conclude there is a difference.

Figure 4A:

There is only 6 wild-type response, not 8 as the authors claimed.

Figure 4B:

The donor has cross-reactive CD4 T cells against SA-R246I pool but the authors state that "none was cross-recognized".

Figure 5:

It is speculated that the authors used expanded cells from the first wave of the COVID pandemic but this needs to be clearly stated.

Figure 7:

Are the percentages of CD38+HLA-DR+ cells in total CD4/CD8 as suggested on the figure axis or in SARS-CoV-2 specific CD4/8 T cells as suggested in the text?

*Reviewer #3 (Recommendations for the authors):*

Methodology section: Additional information on how the peptide specificity was deduced is needed.

Results section: Line 136: Data presentation on T cell polyfunctionality – consider presenting these data in a systematic manner, single cytokines and then the polyfunctionality to avoid discussing data on IFNγ/TNF twice. E.g How do the results on line 136 differ from those shown on lines 156 to 158?

Discussion: authors need to discuss other studies that have presented data on T cell responses to SARS-CoV-2 in HIV-infected individuals. There is no reference to such data in the discussion e.g Sharov IJIAD January 2021, Wang M IJIAD July 2020, Gabriella d'Ettore Medicine 2020, HO et al., JID 2021. Although some are case studies, the authors need to mention and discuss these findings and also discuss the low responses to other proteins besides spike as some previous studies do observe good responses to some of the structural proteins.

Authors observe very low cross-reactivity, other studies have shown reasonable cross-reactivity between SARS-Cov-2 and other human coronaviruses circulating prior to the pandemic. Wouldn't one expect some/better cross-reactivity between SARS-Cov-2 variants?

Figures: correct the y-axis "TNFa" is written as "TFNa".

Line 94: edit to "dominated".

Line 192: clarify the number of participant samples that were included in the long-term culture assays – not completely clear.

Figure 3 y- axis is not clear there is some overlay…

Line 606: There are no arcs around the pie charts. Please include or refer to the colour code on the table besides.

Line 611: include in the legend the structural proteins tested …ie. NS? M? S? N?

Line 700: supplementary figure 1 – indicate which is figure a and b.

Supplementary figure 2: Clarify how many participants were included in this analysis.

---

## [Author Response]

Essential revisions:Reviewer #1 (Recommendations for the authors):In figure 1, the data does not seem internally consistent across pie charts and histograms in panels C and D. Please review and provide an explanation.

We have reviewed the analysis for consistence across the pie charts and histograms as suggested by the reviewer. There are no inconsistencies. The dot plots are based on log10-transformed frequencies of cytokine-producing cells, while the pie charts represent the contribution of each cytokine-producing cell subset in the entire response. For example, (a) if you focus on the IFN^-^IL2^-^TNF^+^ subset in the dot plot of Figure 1C, it has the largest magnitude across all the cytokine-producing cell subsets in the viremic group, and this is also reflected in the pie chart as category 7. (b) if you focus on the IFN^+^IL2^-^TNF^-^ subset in the dot plot of Figure 1C, it has the largest magnitude across all the cytokine-producing cell subsets in the HIV neg group, and this is also reflected in the pie chart as category 3. We have used similar analysis in our previous work *(Mvaya L et al. JCI insight 2021).* Moreover, we have rewritten the Results section on polyfunctionality for clarity.

References to other studies of CoV2 and HIV are incomplete and require greater attention in the introduction and discussion.

More 5 additional papers on HIV and COVID studies relevant this study are now discussed and cited in the second paragraph of the discussion. Additionally, we have discussed 3 more papers that reported different results on cross-recognition to ours and provided a detailed explanation for the different results in paragraph 6 of the revised discussion.

The cross-reactivity data seems to conflict with other studies showing cross-variant reactivity within a person. Please address this in the discussion.

The reviewer raises an important point. We have provided a detailed explanation for the apparent discrepancy between our study and others in the revised discussion as follows. Possible explanation for the apparent discrepancy include, (1) unlike other studies that compared responses to the entire spike protein (*Yu Gao et al. Nat med, 2022*), our cross-recognition studies focused on head-to-head comparisons of wt peptides with corresponding variants peptides containing a lineage defining mutation; (2) We may have picked up fewer cross-reactive responses because we used dual section of IFN-g and TNF-a as a readout for antigen-specific responses, which is more stringent than single cytokine producing cells; (3) we used cultured expansions prior to ICS assays which amplifies the response several folds above background and therefore more specific. Notably, other studies have also reported diminution of responses across variants (*Keton R et al. nature 2022*), which are which consistent with our findings particularly pre-expansion data in Figure 3A and B. Therefore, we recommend that future studies should apply our cultured expansion and the dual cytokine secretion readout to assess cross-recognition among other variants and different vaccine regimens

Reviewer #2 (Recommendations for the authors):With its limited impact on the field, it would have been a more complete manuscript if antibody responses against SARS-CoV-2 had been studied as well and linked with the T cell responses.

The reviewer is correct regarding the importance of T cell and B cells responses. We would like to point out the relationship between B cell responses and total CD8 responses this cohort was examined in the previous study (*Karim F et al. eLife 2021*) even though it was analysed at bulk population level. The study showed that the relationship between T cell and B cell responses was impacted by HIV infection. Unfortunately, we do not have samples to substantiate these findings at antigen-specific level. This limitation is acknowledged in paragraph 8 of the revised discussion.

The first sentence of the abstract claims that HIV infection is one of the major risk factors for severe COVID-19. This statement is not true.

The sentence is revised as follows “In some instances, unsuppressed HIV has been associated with severe COVID-19 disease “. We thank the reviewer for the correction

The sampling time points were from 2 to 4 weeks after the COVID-19 PCR test. The authors need to show the mean time between the PCR test and the analysis of the T cell responses to demonstrate that no difference if biasing the data. Indeed, T cell magnitude can vary between 2 and 4 weeks with earlier sampling showing higher T cell responses.

The mean time between COVID-19 PCR test and T cell analysis is not confounding because our time course measurement showed that the responses peaked between 14 and 30 days after PCR positive test. Moreover, similar time points or longer have been used by other investigators studying T cell responses in South Africa (Keton et al. Cell Host and Microbes 2021, Keeton et al. Nature 2022)

Figure 1B:The authors claim in the text a significantly higher frequency of Spike-specific IFN-γ+ TNF-α+ CD8 T cells between healthy control and HIV viremic but the figure says NS.

We thank the reviewer for spotting the error which is now corrected.

Figures 1C and D:The authors mentioned arcs around the pie chart representing specific cytokine contributions to the pies but there are no arcs in the figure. It would be better to show percentages of total IFN-γ+ cells and TNF-α+ cells within the responding cells.

We thank the reviewer for spotting the error, we have deleted the sentence. However, we do not think it is necessary to include separate graphs of total IFN-γ+ cells and TNF-α+ cells because single cytokine data are shown in the bar graphs and pie charts (Figures 1C and D).

Figure 2:There was no significant difference in the percentage of Spike-specific IFN-γ+ TNF-α+ CD8 T cells between healthy control and HIV viremic in figure 1B. However, there is a significantly higher percentage of the cells in CD8 T cells in this figure. This is confusing. The data from only 4 viremic donors are not strong enough to conclude there is a difference.

Based on the reviewer comment, we realize that the appropriate analysis for Figure 2 is to compare how each individual responded to different SARS-CoV-2 structural proteins, and that is what we now show in the revised figure 2. The text reporting figure 2 is also revised accordingly. We thank the reviewer for comment.

Figure 4A:There is only 6 wild-type response, not 8 as the authors claimed.

Error is now corrected, we thank the reviewer

Figure 4B:The donor has cross-reactive CD4 T cells against SA-R246I pool but the authors state that "none was cross-recognized".

The reviewer is right, the donor represented in Figure 4B had one cross-reactive response, the text is now corrected and the cross-recognized response is highlighted with a red circle in the revised figure 4B.

Figure 5:It is speculated that the authors used expanded cells from the first wave of the COVID pandemic but this needs to be clearly stated.

We now clearly state that the cells where expanded with wildtype peptides from the first wave. The new sentence is highlighted in RED the Results section discussing figure 5. We thank the reviewer for pointing out this omission

Figure 7:Are the percentages of CD38+HLA-DR+ cells in total CD4/CD8 as suggested on the figure axis or in SARS-CoV-2 specific CD4/8 T cells as suggested in the text?

Responses in Figure 7 are for the total CD4/CD8 as indicated in the figures. We have removed the phrase “SARS-CoV-2 specific” in the Results section discussing figure 7. We thank the reviewer for this important observation.

Reviewer #3 (Recommendations for the authors):Methodology section: Additional information on how the peptide specificity was deduced is needed.

Peptide specificity was determined by cytokine secretion following in vitro stimulation with a particular peptide. This is now explained in the ex vivo cultured expansion methods section as follows: “Peptides that induced IFN-γ / TNF-α dual production above background (No stimulation control) were deemed reactive. Meaning that, the expanded cells had significant numbers of cells that were (reactive) specific for that particular peptide”. The new text is highlighted in red

Results section: Line 136: Data presentation on T cell polyfunctionality – consider presenting these data in a systematic manner, single cytokines and then the polyfunctionality to avoid discussing data on IFNγ/TNF twice. E.g How do the results on line 136 differ from those shown on lines 156 to 158?

The reviewer is right the text in line text in line 156 was a repetition of line 136. We have rewritten the whole section for clarity.

Discussion: authors need to discuss other studies that have presented data on T cell responses to SARS-CoV-2 in HIV-infected individuals. There is no reference to such data in the discussion e.g Sharov IJIAD January 2021, Wang M IJIAD July 2020, Gabriella d'Ettore Medicine 2020, HO et al., JID 2021. Although some are case studies, the authors need to mention and discuss these findings and also discuss the low responses to other proteins besides spike as some previous studies do observe good responses to some of the structural proteins.

All the recommended papers and more are now included in the revised discussion. We thank the reviewer for the instructive comment.

Authors observe very low cross-reactivity, other studies have shown reasonable cross-reactivity between SARS-Cov-2 and other human coronaviruses circulating prior to the pandemic. Wouldn't one expect some/better cross-reactivity between SARS-Cov-2 variants?

We agree with the reviewer that some studies have shown more cross-reactivity than what we report. The discrepancy could be due to the differences in methods used. Here we used cultured expansion which amplifies the signal. The differences between studies could also be due to differences in populations studied, variants or vaccines. Importantly, our data show that immune compromised individuals have poor cross-reactive responses. All these points are raised in the revised discussion in paragraph 6.

Figures: correct the y-axis "TNFa" is written as "TFNa".

The typo is corrected in all the figures, we thank the reviewer

Line 94: edit to "dominated".

Line 94 is now edited to dominated.

Line 192: clarify the number of participant samples that were included in the long-term culture assays – not completely clear.

The number of participants used in the long-term cultures is 12, 4 for each category. This is now stated (highlighted in red) in the Results section reporting figure 5 results.

Figure 3 y- axis is not clear there is some overlay.

Figure 3Y is now fixed

Line 606: There are no arcs around the pie charts. Please include or refer to the colour code on the table besides.

We have removed the sentence. We thank the reviewer for pointing out the error

Line 611: include in the legend the structural proteins tested …ie. NS? M? S? N?

The structural proteins tested are now included in the Figure 2 legend

Line 700: supplementary figure 1 – indicate which is figure a and b.

In suppl Figure 1 Figure A and B are now indicated.

Supplementary figure 2: Clarify how many participants were included in this analysis.

The number of participants included are indicated in supplementary Figure 2 graphs.